# No Evidence for Classic Thrombotic Microangiopathy in COVID-19

**DOI:** 10.3390/jcm10040671

**Published:** 2021-02-09

**Authors:** Tanja Falter, Heidi Rossmann, Philipp Menge, Jan Goetje, Steffen Groenwoldt, Arndt Weinmann, Visvakanth Sivanathan, Andreas Schulz, Niels A.W. Lemmermann, Sven Danckwardt, Karl J. Lackner, Peter R. Galle, Inge Scharrer, Bernhard Lämmle, Martin F. Sprinzl

**Affiliations:** 1Institute of Clinical Chemistry and Laboratory medicine, University Medical Center of the Johannes Gutenberg University Mainz, 55131 Mainz, Germany; tanja.falter@unimedizin-mainz.de (T.F.); heidi.rossmann@unimedizin-mainz.de (H.R.); sven.danckwardt@unimedizin-mainz.de (S.D.); karl.lackner@unimedizin-mainz.de (K.J.L.); 2Medical Department I, University Medical Center of the Johannes Gutenberg University Mainz, 55131 Mainz, Germany; pmenge@students.uni-mainz.de (P.M.); jan.goetje@gmx.de (J.G.); sgroenwo@students.uni-mainz.de (S.G.); arndt.weinmann@unimedizin-mainz.de (A.W.); visvakanth.sivanathan@unimedizin-mainz.de (V.S.); galle@uni-mainz.de (P.R.G.); 3Clinical Registry Unit, University Medical Center of the Johannes Gutenberg University Mainz, 55131 Mainz, Germany; 4Center for Thrombosis and Hemostasis, University Medical Center of the Johannes Gutenberg University Mainz, 55131 Mainz, Germany; andreas.schulz@unimedizin-mainz.de (A.S.); inge.scharrer@unimedizin-mainz.de (I.S.); bernhard.laemmle@uni-mainz.de (B.L.); 5Institute of Virology, University Medical Center of the Johannes Gutenberg University Mainz, 55131 Mainz, Germany; lemmermann@uni-mainz.de; 6Department of Hematology and Central Hematology Laboratory, Inselspital, Bern University Hospital, University of Bern, CH 3010 Bern, Switzerland; 7Haemostasis Research Unit, University College London, London WC1E 6BT, UK

**Keywords:** coronavirus disease, COVID-19, ADAMTS13, microangiopathy, disseminated intravascular coagulation

## Abstract

Background: Coronavirus disease-2019 (COVID-19) triggers systemic infection with involvement of the respiratory tract. There are some patients developing haemostatic abnormalities during their infection with a considerably increased risk of death. Materials and Methods: Patients (*n* = 85) with SARS-CoV-2 infection attending the University Medical Center, Mainz, from 3 March to 15 May 2020 were retrospectively included in this study. Data regarding demography, clinical features, treatment and laboratory parameters were analyzed. Twenty patients were excluded for assessment of disseminated intravascular coagulation (DIC) and thrombotic microangiopathy (TMA) due to lack of laboratory data. Results: COVID-19 patients (*n* = 65) were investigated, 19 with uncomplicated, 29 with complicated, and 17 with critical course; nine (13.8%) died. Seven patients showed overt DIC according to the ISTH criteria. The fibrinogen levels dropped significantly in these patients, although not below 100 mg/dl. Hallmarks of TMA, such as thrombocytopenia and microangiopathic haemolytic anaemia, were not detected in any of our COVID-19 patients. ADAMTS13 activity was mildly to moderately reduced in 4/22 patients, all having strongly elevated procalcitonin levels. Conclusion: DIC occurred in 7/65 COVID-19 patients but fibrinogen and platelet consumption were compensated in almost all. ADAMTS13 assays excluded TTP and hallmarks of classic TMA were absent in all investigated patients. We hypothesize that the lacking erythrocyte fragmentation and only mild platelet consumption in severe COVID-19 are due to a microangiopathy predominantly localized to the alveolar microcirculation with a low blood pressure gradient.

## 1. Introduction

Patient age, male sex and pre-existing comorbidities are the major determinants of clinical severity and outcome of coronavirus disease-2019 (COVID-19) caused by Severe Acute Respiratory Syndrome-Coronavirus-2 (SARS-CoV-2) [1,2,3]. Some haemostasis parameters have been associated with poor outcome [4,5,6] and D-dimer values >2 µg/mL predicted mortality among hospitalized COVID-19 patients [7]. These findings led to the recommendation to monitor haemostasis parameters in COVID-19 patients [8]. In line with laboratory surrogates of activated coagulation, COVID-19 is associated with increased rates of thromboembolic events in 15% up to 69% [9,10,11,12,13]. Autopsy studies have confirmed that large vessel thromboembolic events and microthrombosis contribute to structural lung damage and respiratory failure [14,15,16]. Disseminated intravascular coagulation (DIC), as identified by a diagnostic DIC score, was found in 15/21 (71.4%) patients with fatal outcome but only 1/162 (0.6%) survivors [4]. DIC potentially evolves from endothelial activation or endothelial damage triggered by SARS-CoV-2 infection and subsequent consumption of plasmatic coagulation factors during COVID-19 [17,18,19]. DIC causes microvascular thrombosis, subsequent tissue malperfusion, and eventually drives multi organ failure. The clinical presentation of DIC during COVID-19 may have a similar appearance as thrombotic microangiopathies (TMAs) [20]. Classic TMAs, including thrombotic thrombocytopenic purpura (TTP), haemolytic uremic syndrome (HUS) and a series of other TMAs, describe an etiologically very heterogeneous group of conditions. TMAs are characterized by microvascular endothelial damage with increased release of von Willebrand factor (VWF) and widespread arteriolar and capillary thrombosis leading to the diagnostic hallmarks of consumptive thrombocytopenia and microangiopathic haemolytic anaemia (MAHA) with schistocytes in the blood smear [21,22].

A massive release of VWF from the endothelial cells, as it occurs in severe inflammatory states and systemic infections, can lead to a mild decrease of VWF-cleaving protease, a disintegrin and metalloprotease with thrombospondin type 1 motif 13 (ADAMTS13) [23]. Whether the resulting VWF/ADAMTS13 dysbalance contributes to the pathophysiology of certain TMAs in a similar way as in classic TTP characterized by a very severe deficiency of ADAMTS13 activity (<5–10% of normal) remains unclear.

We investigated clinical and laboratory patterns in this observational study to understand more about the underlying coagulopathy during COVID-19. In particular, we focused on clinical and laboratory features of DIC and classic TMAs.

## 2. Materials and Methods

Patients (*n* = 85) with confirmed SARS-CoV-2 infection who were seen at the University Medical Center, Mainz, Germany, between 3 March and 15 May 2020 were assessed in this observational study. Patient characteristics and laboratory findings were reviewed retrospectively through the electronic hospital information systems (i.s.h.med^®^, SAP, Weinheim Germany, Nexus Swisslab, Berlin, Germany). The retrospective study was approved by German law [Landeskrankenhausgesetz §36 and §37] in accordance with the Declaration of Helsinki and by the local Ethics Committee of “Landesärztekammer Rheinland-Pfalz” (reference numbers: 2020-14988_2).

Severity of COVID-19 was classified by respiratory function into an uncomplicated, complicated, and critical clinical course. Patients with uncomplicated disease required neither monitoring nor oxygen supplementation, whereas patients affected by complicated COVID-19 were in need for oxygen supplementation and critically ill COVID-19 patients needed invasive ventilation. The categorization into the individual COVID-19 severity stages was done retrospectively based on the clinical course during hospitalization.

SARS-CoV-2 infection was confirmed by polymerase chain reaction (PCR) from respiratory samples, employing a PCR kit specific for SARS CoV-2 (Altona Diagnostics GmbH, Hamburg, Germany). All other laboratory assays were performed in the accredited (DIN-ISO 15.189) Institute of Clinical Chemistry and Laboratory Medicine of the University Medical Center, Mainz. Renal injury was assessed based on Acute Kidney Injury Network (AKIN) criteria [24,25]. Presence of DIC was determined according to ISTH guidelines by the DIC score, incorporating platelet count, D-dimer, INR and fibrinogen level [26]. D-Dimer, derived fibrinogen and prothrombin time (PT/INR) were performed on ACL TOP 750 instruments (Instrumentation Laboratory Company, IL, Bedford, MA, USA) using IL reagents (HemosIL D-Dimer HS 500 and HemosIL RecombiPlasTin 2G) and following the manufacturer’s instructions. ADAMTS13 activity was examined by the fluorescence resonance energy transfer system (FRETS-VWF73) method [27] modified according to Kremer-Hovinga et al. [28].

Statistical analyses employed SPSS version 22.0 (IBM GmbH, Ehningen, Germany). Descriptive statistics included frequency, mean, standard deviation, median, interquartile range (IQR), minimum and maximum. Explorative group comparisons were performed by *t*-test or Mann–Whitney-U-test for continuous variables and by Chi-squared test or Fisher’s exact test for categorical variables, accordingly. *p*-values are two tailed, and *p* values < 0.05 were considered statistically significant.

## 3. Results

### 3.1. Patient Characteristics

A total of 85 patients with proven SARS-CoV-2 infection were seen between 3 March and 15 May 2020. For the assessment of haemostatic alterations, 20 patients were excluded due to insufficient laboratory data (Table 1 and Appendix A).

Sixty-three/65 (97%) COVID-19 patients were hospitalized and 20/65 (31%) had to be treated in intensive care unit. COVID-19 patients were predominantly male (63%) and had a median age of 69 (IQR 57–79, range 22–86) years. Older adults (age > 65 years) accounted for 37/65 (57%). The most common underlying comorbidities were arterial hypertension, cardiovascular disease and diabetes mellitus. Obesity (BMI ≥ 30 kg/m^2^) was observed in 23/62 (37%) and a history of venous thromboembolic events before COVID-19 in 4/65 (6%). The clinical course of COVID-19 was uncomplicated in 19 (29%), complicated in 29 (45%) and critical in 17 (26%). The overall mortality rate was 14% (9/65), reaching 35% (6/17) in critical COVID-19. Invasive ventilation of 17 critically ill patients was performed over a period of 21 (median, IQR 7–30) days. Prophylactic or therapeutic anticoagulation was used in 43/46 (94%) patients with complicated or critical COVID-19 manifestation and in 14/19 (74%) of the uncomplicated cases (Table 1).

### 3.2. COVID-19-Associated Laboratory Parameters and Organ Damage

Elevated creatinine values were observed in 38/64 (59%) throughout the course of COVID-19 (Table 2 and Appendix A). According to the Classification for Acute Kidney Injury (AKIN) patients with COVID-19 developed a new onset of renal injury (AKIN 1) and renal failure (AKIN 3) in 33/65 (5%) and 15/65 (23%), respectively (Table 1). Renal replacement therapy was eventually initiated in 10/65 (15.4%) patients (Table 1).

Elevated troponin I (>24 pg/mL) indicative of myocardial injury was observed in 27/61 (44%) of all COVID-19 patients (Table 2 and Appendix A). In patients with myocardial injury, 9/27 (33%) had arrhythmic events, 2/27 had a myocarditis and 3/27 (11%) were diagnosed with acute coronary syndrome (Table 1).

Significant increases (>5 times the upper limit of normal) in AST and ALT were observed in 17/63 (27%) and 9/64 (14%), respectively. Impaired liver function as indicated by hyperbilirubinemia (total bilirubin >1.2 mg/dL) and pronounced hypoalbuminemia (serum albumin <28 g/L) occurred in 17/64 (27%) and 31/58 (53%), respectively (Table 2 and Appendix A).

### 3.3. COVID-19 Associated Haemostatic Alterations

#### 3.3.1. Thromboembolic Events during COVID-19

Among our COVID-19 patients, acute new-onset thromboembolic (TE) events were observed in 4/65 (6%) (Table 1). These TE events predominantly affected patients with critical COVID-19 (3/17, 18%) and only 1/19 patients (5%) with uncomplicated COVID-19. The TE events during critical COVID-19 included two patients with venous thrombosis associated with central vein catheters and one with acute arterial mesenterial infarction. The patient with uncomplicated COVID-19 developed a segmental pulmonary embolism. Twenty-seven patients had cardiac troponin I elevation (Appendix A) and three of them required interventional coronary angioplasty for acute coronary syndrome (Table 1).

As shown in Table 1, 57/65 patients (88%) were treated with prophylactic or therapeutic doses of LMW-heparin from the time of hospitalization. In particular, all patients who developed thromboembolic events during COVID-19 received therapeutic doses of LMW-heparin. One patient had been under rivaroxaban because of deep vein thrombosis and lung embolism that had occurred prior to COVID-19. No major bleeding event was observed during hospitalization and heparin-induced thrombocytopenia did not occur.

#### 3.3.2. D-Dimers, Fibrinogen, INR, Platelet Count

Elevation of INR and D-dimers along with COVID-19 severity was evident (Table 2 and Appendix A and Figure 1). Consequently, the highest median INR (1.6, IQR 1.5–2.2, range 1.4–10.6) and highest median D-dimer (6.16 mg/L, IQR 4.36–17.23 mg/L, range 1.22–43.72 mg/L) was reached among critically ill COVID-19 patients. D-dimer concentrations > 2 mg/L, which had been associated with adverse outcome [7], were observed in 25/65 (40%) of the entire cohort and in 13/17 (81%) patients with critical COVID-19 (Table 2). In contrast, platelet counts and fibrinogen levels did not significantly differ between patient subgroups of increasing COVID-19 severity (Table 2 and Appendix A and Figure 1).

Fibrinogen levels in COVID-19 patients were elevated in 49/58 (84%) at baseline (median 531, IQR 465–660, range 275–933 mg/dL). Particularly all patients (17/17) with critical COVID-19 presented with elevated baseline fibrinogen levels (median 714, IQR 613–855, range 466–933 mg/dL). During the course of COVID-19 maximum fibrinogen concentrations were above normal in the entire COVID-19 cohort (Table 2 and Appendix A) and remained elevated for several days in most patients

#### 3.3.3. Patients with DIC

Overt disseminated intravascular coagulation (DIC), as defined by a DIC Score ≥5 points (26), was present in 7/65 (11%) patients (Table 3 and Appendix A, Figure 2).

The dynamics of laboratory values included into the DIC score are shown for the seven patients who developed DIC over the course of hospitalisation (Figure 2). The DIC score became positive after a median of 21 (range 6.5–40) days post hospital admission. DIC diagnosis was mainly based on moderately (0.5–2.0 mg/L) or extremely (>2.0 mg/L) elevated D-dimers and an INR above 1.7. Patients with positive DIC score included only four patients (Pat.ID 24/ 50/ 74/ 79) with thrombocytopenia (platelets < 100/nL) (Figure 2). Fibrinogen levels were elevated in all seven patients at hospitalisation, followed by a decrease in fibrinogen levels over time (Figure 2). However, fibrinogen remained ≥100 mg/dl in all seven patients meeting the criteria for DIC (Table 3 and Figure 2). DIC was only diagnosed in male patients with complicated or critical COVID-19, three patients died (Table 1 and Figure 2). DIC patients had more pronounced LDH, total bilirubin and creatinine elevations as compared to the non-DIC group (Table 2 and Appendix A). Two patients received low molecular weight heparin in prophylactic doses, three patients in therapeutic doses and one patient continued rivaroxaban introduced prior to COVID-19 infection (Table 1).

#### 3.3.4. Markers of Endothelial Damage and Thrombotic Microangiopathy

Some authors reported that severe COVID-19 may cause thrombotic microangiopathy (TMA) [20,29,30,31]. Therefore, we examined patients for microangiopathic haemolytic anaemia (MAHA) and consumptive thrombocytopenia as hallmarks of classic TMA [21].

Anaemia using a haemoglobin cut-off for males (<13.5 g/dL) and females (<12.0 g/dL) was observed in 52/65 (80%) of our COVID-19 patients (Appendix A). Haemoglobin reduction paralleled the severity of COVID-19, leading to anaemia in 23/29 (79%) patients with complicated COVID-19 and in 17/17 (100%) patients with critical COVID-19. Despite elevated LDH levels in 60/63 (95%) patients (Appendix A) being compatible with haemolysis, available haptoglobin values from 22 patients across all COVID-19 stages were all normal or elevated (Table 4). In addition, schistocytes were either absent in the peripheral blood smear of 20 investigated patients or only minimally elevated (i.e., 5‰ and 9‰) in two. Based on these findings MAHA with intravascular haemolysis was not observed in this cohort (Table 4).

Platelet counts were <150/nL in 22/65 (34%) without clear relation to disease severity (Appendix A). However, thrombocytopenia was generally mild (platelets 100 – <150/nL) in 15/65 (23%) or moderate (platelets 50 – <100/nL) in 4/65 (6%) (Appendix A). Severe thrombocytopenia (platelets <50/nL) was found in only three (5%) patients of whom one had received cytotoxic chemotherapy.

Finally, ADAMTS13 was mostly in the normal range and only 4/22 tested showed reduced ADAMTS13 activity values (<50%) with a minimum ADAMTS13 activity of 17.8% (Table 4). These mild or moderately decreased ADAMTS13 activity values excluded thrombotic thrombocytopenic purpura in any patient. Most prominent finding in these 22 patients, comprehensively tested for the presence of TMA, were the elevated VWF activity (median 329%, IQR 195 – >390%) and antigen (median 232%, IQR 219–498%) levels (Table 4). The VWF antigen/ ADAMTS13 activity ratio was elevated in 21/21 (100%) patients tested (median 3.4, IQR 2.6–7.7, range 2.1–33.4) (Table 4).

It is remarkable that in the four patients with mild to moderately reduced ADAMTS13 activity (18–48%) by far the highest procalcitonin (PCT) values were found. Bacterial coinfections were confirmed in two cases by *Escherichia coli* and *Staph epidermidis* isolates from the lower respiratory system and blood stream, respectively.

In sum, there were no diagnostic clues for classic TMA (lacking schistocytes, normal or elevated haptoglobin levels, no severe thrombocytopenia) in any of our patients whereas the consistently high VWF levels were compatible with an endothelial activation and/or damage.

## 4. Discussion

We present here our single-centre cohort of all consecutive, retrospectively included patients diagnosed with COVID-19 at the University Medical Center Mainz during the first wave of the pandemic from 3 March to 15 May 2020. Our report focuses on the clinical and laboratory abnormalities related to thrombosis and haemostasis. Multiple scientific publications pointing to the high thrombotic risk in COVID-19 [9,10,11,12,13,32], trying to understand its pathophysiology [33,34,35], suggesting the prognostic value of haemostatic laboratory parameters [4,36,37] and proposing prophylactic and/or therapeutic measures to improve the outcome have been provided [11,38].

From 65 of the 85 registered COVID-19 patients sufficient laboratory data were available to delineate the frequency and extent of haemostatic abnormalities (Table 1). Nine of the patients died, 3 of 29 with complicated and 6 of 17 with critical disease. No autopsies were performed. The incidence of clinically manifest thromboembolic events was rather low and included three venous TE, one mesenteric arterial infarction and three acute coronary syndromes needing percutaneous coronary intervention (Table 1), which is substantially lower than reported by several authors [12,13]. Whereas initially published cohorts from China had not received TE prophylaxis [9], other series of patients had a cumulative incidence of TE events up to about 15–60% (dependent on COVID-19 severity and length of hospitalisation) despite prophylactic or even higher-than-prophylactic doses of LMWH given [13]. Major bleeding events in patients with or without anticoagulation were generally rare [39].

Routine haemostatic laboratory parameters showed strongly elevated fibrinogen and D-dimer levels, the latter being exceedingly high in those with critical disease (Table 2, Figure 1). Prothrombin times expressed as INR were mildly elevated in critical patients and platelet counts were mostly normal and sometimes subnormal independent of COVID-19 severity. Calculating the DIC score [26] taking into account the highest D-dimer and INR and the lowest fibrinogen and platelet count values showed seven of 65 patients having a score ≥5 signalling overt DIC (Table 3 and Appendix A). The course of laboratory parameters in these seven individual patients shows a notable difference of their “DIC pattern” as compared to “typical DIC” associated with bacterial sepsis, obstetric complications and other inflammatory conditions [40,41]. Six of our DIC patients showed a mild drop of fibrinogen (but none <100 mg/dL), platelet count fell below 50/nL in only two, and none of the seven showed any abnormal bleeding tendency. These data resemble observations by other groups [20,37] and new designations for the COVID-19 associated haemostatic disturbances have been proposed, i.e., “COVID-19 associated coagulopathy (CAC)” (18) or “Pulmonary intravascular coagulopathy (PIC)” [42] to stress the discrepancy to “classic DIC” [43]. Experts in this field have discussed the current evidence and suggested that CAC during severe COVD-19 should be considered as a prothrombotic phenotype of DIC [44].

Other investigators have labelled the coagulopathy in COVID-19 as thrombotic microangiopathy (TMA) [30,31,45,46,47,48]. Nevertheless, the hallmarks of classic TMA, consumptive thrombocytopenia and microangiopathic haemolysis with erythrocyte fragmentation resulting in schistocytes in the peripheral blood smear [21], have neither been found in any of our 22 patients subjected to detailed investigation for TMA (Table 4) nor in most studies from other investigators [49,50]. Mildly or moderately decreased ADAMTS13 activity (18–48% of normal in 4 of the 22 patients, the other 18 displaying normal activity >50%) (Table 4) as well as normal [50] or subnormal to normal levels [30,45,49] in several previously described patients clearly ruled out thrombotic thrombocytopenic purpura (TTP), which is characterized by <10% (and indeed often <1%) ADAMTS13 activity [51]. Most notable findings in the 22 patients were the highly elevated values of VWF activity and antigen and the increased ratio of VWF:antigen/ADAMTS13 activity (Table 4). Whether this imbalance between high VWF concentrations and low levels of its size-regulating protease in our patients, features that have also been described in patients with severe sepsis or septic shock [23,52], is pathophysiologically relevant in COVID-19 remains questionable, especially in the absence of consumptive thrombocytopenia. Henry et al. reported that a decreasing ratio of ADAMTS13:act/ VWF:Ag in 52 COVID-19 patients at presentation to the emergency room was predictive for the development of acute kidney injury and a severe form of COVID-19 [48]. However, the decreased ratio of ADAMTS13:act/VWF:Ag in 12 of 52 patients, which did not correlate with platelet count, was mainly due to high VWF:Ag levels. This may, in turn, primarily reflect an augmented VWF release caused by endothelial injury [48]. It is well possible that the highly elevated FVIII:C levels, paralleling the increased VWF values in COVID-19 [49,50] are equally or more relevant in mediating a prothrombotic effect. The explanation for the consistently and often massively elevated VWF values is likely explained by endothelial activation and damage [14,17,50], the latter caused by direct endothelial invasion by the SARS-CoV-2 [14,53,54]. The autopsy study by Ackermann and colleagues comparing the lungs of seven patients who died from COVID-19 with those of seven who died from acute respiratory distress syndrome caused by influenza A infection and 10 age-matched control lungs showed distinctive morphologic vascular features in COVID-19. These findings included severe endothelial injury, widespread alveolar capillary fibrinous microthrombi (9-times more prevalent than in influenza-infected lungs) and marked features of (mainly intussusceptive) angiogenesis with formation of new vessels growing into the lumen of existing vessels [14]. Beigee et al. described diffuse alveolar damage and thrombotic microangiopathies in lung biopsies of 31 patients who had died from COVID-19 [16]. Moreover, other autopsy studies confirmed extensive fibrinous microthrombi in the lungs [53,55] and also in some skin lesions [53], from patients succumbed to severe COVID-19. The latter study demonstrated, in addition, significant microvascular deposition of complement activation products, C5b-9, C4d and mannose binding lectin-associated serine protease 2, in colocalization with SARS-Cov-2, hinting at systemic activation of the alternative and lectin-mediated complement pathways [53]. Thus, these studies provide unequivocal proof of “thrombotic microangiopathy”, mainly in the lungs. A study of 50 children with COVID-19 including 18 with a multisystem inflammatory syndrome in children (MIS-C) suggested the presence of TMA in 17 of 19 with complete laboratory evaluation. Many showed evidence of systemic complement activation, i.e., increased sC5b-9 levels [47]. Furthermore, first studies describe the successful use of complement inhibitors in COVID-19 patients [56].

Whether the (almost uniformly) lacking laboratory hallmarks of TMA in our COVID-19 patients and those described in other cohorts [49,50] is explained by the predominantly pulmonary microangiopathy in COVID-19 remains to be investigated. While both VWF-mediated platelet adhesion and aggregation mainly in the microcirculation of brain, kidney and heart as seen in TTP [51] and fibrin microthrombi located predominantly in the kidney as seen in HUS [21,22], will result in erythrocyte fragmentation in the partially occluded microcirculation with the high blood pressure gradient in the arterial circulation, it may be hypothesized that the “COVID-19-associated thrombotic microangiopathy that is often restricted to the pulmonary microcirculation will not produce schistocytes owing to the much lower blood pressure gradient”.

### Limitations of the Study

Our study has several limitations. Due to the retrospective approach, relevant parameters were not monitored throughout the course of COVID-19. In particular, we did not assess markers of complement activation, which may play a pathophysiologic role in the prothrombotic alterations of COVID-19 [53] and inhibition of complement activation may be targeted therapeutically [56]. In addition, several other tests, such as viscoelastic methods to study clot formation and fibrinolysis were not performed in this COVID-19 cohort. Whether these latter methods could be useful to identify the thrombotic risk in individual patients and/or to guide antithrombotic treatment remains to be further investigated [57].

Another limitation is a bias regarding the severity of COVID-19. Mostly severely symptomatic patients presented at our University Medical Center and were admitted as inpatients and laboratory data on haemostatic alterations were not available from all patients. Therefore, our observational study is merely hypothesis-generating.

## 5. Conclusions

In sum, our observational data on the haemostaseologic abnormalities in real world adult COVID-19 patients admitted to a single academic centre add results suggesting that the haemostatic alterations in severe COVID-19 are not fully fitting into the established categories of DIC and classic TMA. Further pathophysiologic research is needed to provide clinically useful targets for intervention to improve patient outcome.

## Figures and Tables

**Figure 1 jcm-10-00671-f001:**
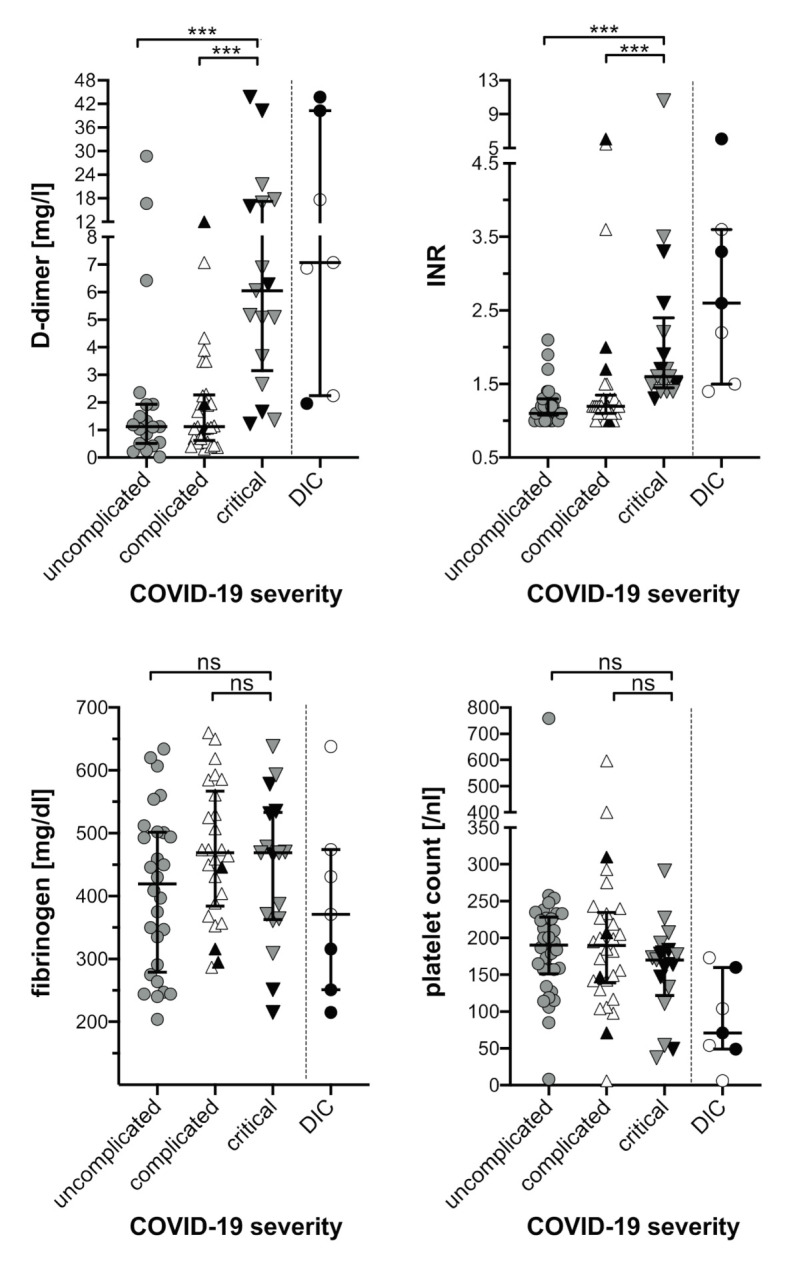
Haemostasis parameters of COVID-19 patients. The highest values for D-dimer and INR as well as the lowest values for fibrinogen and platelet count are plotted against the severity of COVID-19 disease and the occurrence of DIC as indicated. Black symbols represent laboratory values of deceased COVID-19 patients. Median and interquartile range are provided. Comparisons between subgroups were based on Mann–Whitney U Test (ns, not significant; *** *p* < 0.001). COVID-19, coronavirus disease-19; DIC, disseminated intravascular coagulation; INR, international normalized ratio.

**Figure 2 jcm-10-00671-f002:**
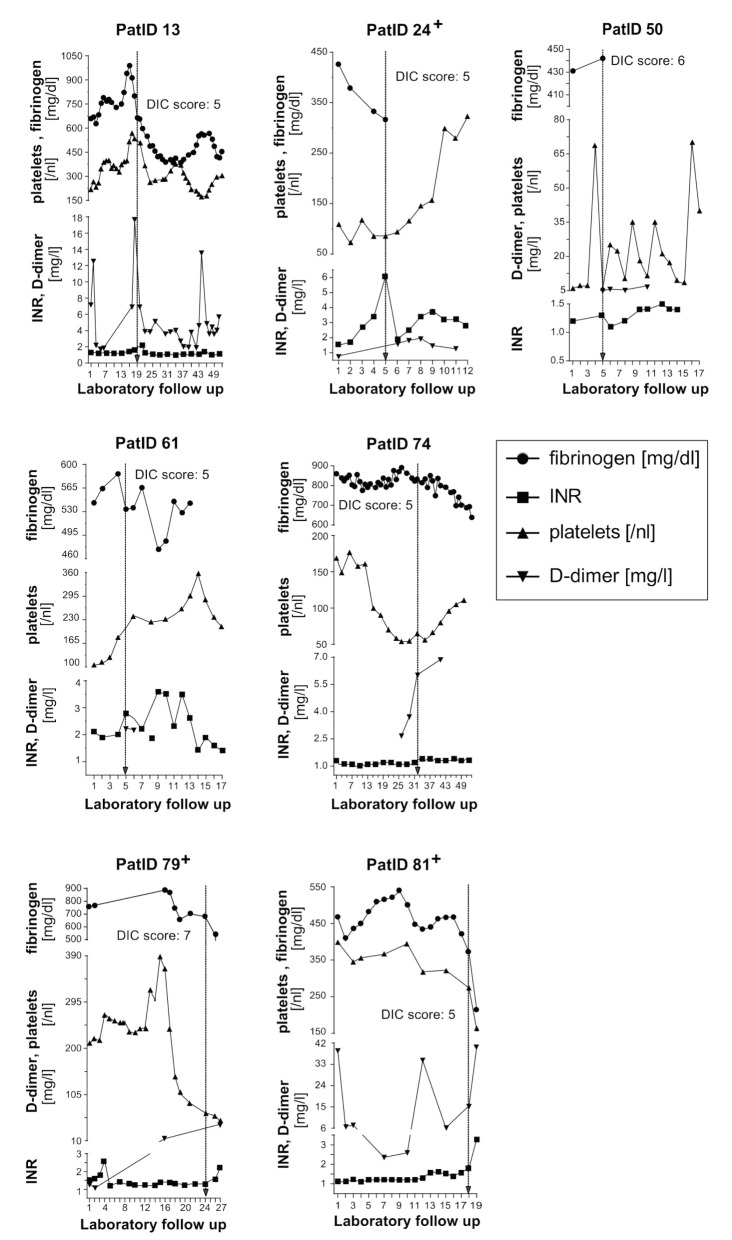
Time course of fibrinogen, INR, platelet count and D-dimer values in seven patients who developed DIC during COVID-19. The engraved arrow indicates the time of DIC manifestation as defined by the corresponding DIC score. Three patients with DIC died as indicated (+). DIC, disseminated intravascular coagulation; INR, international normalized ratio; PatID, patient identification number.

**Table 1 jcm-10-00671-t001:** Characteristics of 65 patients with COVID-19 and analyzed for haemostatic abnormalities.

Characteristics	Total	Uncomplicated COVID-19	Complicated COVID-19	Critical COVID-19	DIC ^a^
Number	65	19	29	17	7
Age (years) (IQR)	69 (57–79)	64 (39–79)	74 (60–81)	66 (53–73)	76 (55–80)
Sex (male/female)	41/24 (63.1/36.9%)	10/9 (52.6/47.4%)	17/12 (58.6/41.4%)	14/3 (82.4/17.6%)	7/0 (100/0.0%)
BMI (kg/m^2^) (IQR) ^b^	27.2 (24.2–33.6)	26.7 (22.5–32.6)	29.4 (24.4–33.6)	27.0 (25.7–33.5)	27.0 (24.7–29.4)
**Preexisting comorbidities**					
Arterial Hypertension	38 (58.5%)	9 (47.4%)	9 (31%)	10 (58.8%)	6 (85.6%)
Diabetes mellitus	14 (21.5%)	1 (5.3%)	8 (27.6%)	5 (29.4%)	2 (28.6%)
Obesity (BMI ≥30 kg/m^2^) ^b^	23/62 (37.1%)	5/16 (31.1%)	13/29 (44.8%)	5/17 (29.4%)	1/7 (14.3%)
Chronic respiratory disease	11 (16.9%)	2 (10.5%)	6 (20.7%)	3 (17.6%)	1 (14.3%)
Cardiovascular disease	17 (26.2%)	5 (26.3%)	8 (27.6%)	4 (23.5%)	3 (42.9%)
Cerebrovascular disease	11 (16.9%)	2 (10.5%)	3 (10.3%)	6 (35.3%)	1 (14.3%)
Terminal renal insufficiency	2 (3.1%)	1 (5.3%)	1 (3.4%)	0 (0.0%)	0 (0.0%)
Venous thromboembolic history	4 (6.2%)	0 (0.0%)	1 (3.4%)	3 (17.6%)	1 (14.3%)
Pulmonary Embolism	1 (1.5%)	0 (0.0%)	1 (3.4%)	0 (0.0%)	0 (0.0%)
Deep vein thrombosis	4 (6.2%)	0 (0.0%)	1 (3.4%)	3 (17.6%)	1 (14.3%)
**Preexisting anticoagulation**					
DOAC	1 (1.5%)	0 (0.0%)	1 (3.4%)	0 (0.0%)	1(14.3%)
Vitamin K antagonist	1 (1.5%)	0 (0.0%)	0 (0.0%)	1 (5.9%)	0 (0.0%)
Platelet aggregation inhibitors	19 (29.2%)	8 (42.1%)	7 (24.1%)	4 (23.5%)	0 (0.0%)
**Clinical complications during COVID-19 infection**					
Renal failure	18 (27.7%)	0 (0.0%)	5 (17.2%)	13 (76.5%)	4 (57.1%)
AKIN 1	3 (4.6%)	0 (0.0%)	2 (6.9%)	1 (5.9%)	0 (0.0%)
AKIN 3	15 (23.1%)	0 (0.0%)	3 (10.3%)	12 (70.6%)	4 (57.1%)
Thromboembolic events	4 (6.2%)	1 (5.3%)	0 (0.0%)	3 (17.6%)	1 (14.3%)
Acute coronary syndrome	3 (4.6%)	0 (0.0%)	2 (6.9%)	1 (5.9%)	1 (14.3%)
Myocarditis	2 (3.1%)	2 (3.1%)	0 (0.0%)	0 (0.0%)	0 (0.0%)
**Medical care during COVID-19 infection**					
Hospitalized	63 (96.9%)	17 (89.5%)	29 (100%)	17 (100%)	7 (100%)
Intensive care	20 (30.8%)	0 (0.0%)	3 (10.3%)	17 (100%)	4 (57.1%)
Oxygen supplementation	46 (70.8%)	-	29 (100%)	17 (100%)	7 (100%)
Invasive ventilation	17 (26.2%)	-	-	17 (100%)	4 (57.1%)
Renal replacement therapy	10 (15.4%)	0 (0.0%)	0 (0.0%)	10 (58.8%)	3 (42.9%)
**Anticoagulation during COVID-19 infection**					
None	7 (10.8%)	5 (26.3%)	2 (6.9%)	0 (0.0%)	1 (14.3%)
Prophylactic dose LMWH	43 (66.2%)	13 (68.4%)	25 (86.2%)	5 (29.4%)	2 (28.6%)
Therapeutic dose LMWH	14 (21.5%)	1 (5.3%)	1 (3.4%)	12 (70.6%)	3 (42.9%)
DOAC	1 (1.5%)	0 (0.0%)	1 (3.4%)	0 (0.0%)	1 (14.3%)
**Clinical outcome of COVID-19 infection**					
Uncomplicated	19 (29.2%)	19 (100%)	-	-	0 (0.0%)
Complicated	29 (44.6%)	-	29 (100%)	-	3 (42.9%)
Critical	17 (26.2%)	-	-	17 (100%)	4 (57.1%)
Deceased	9 (13.8%)	0 (0%)	3 (10.3%)	6 (35.3%)	3 (42.9%)

Patient characteristics are presented as median (interquartile range) or number (%). Explorative comparisons of patient subgroups and corresponding *p*-values are provided in Appendix A. AKIN, AKIN Classification for Acute Kidney Injury; BMI, body mass index; COVID-19, coronavirus disease-2019; DOAC, direct oral anticoagulants; DIC, disseminated intravascular coagulation; IQR, interquartile range; LMWH, low molecular weight heparin. ^a^ Patients with DIC are a subset of patients with complicated and critical COVID-19. ^b^ BMI and obesity could only be determined in 62 of 65 patients due to missing anthropometric data.

**Table 2 jcm-10-00671-t002:** Laboratory analyses of 65 patients with COVID-19 analyzed for haemostatic abnormalities.

Parameter	Total		Uncomplicated COVID-19		Complicated COVID-19		Critical COVID-19		DIC ^a^	
**Number of Patients**	**65**	#	**19**	#	**29**	#	**17**	#	**7**	#
LDH^max^ (U/L)	507 (381–705)	63	421 (345–476)	17	495 (367–593)	29	705 (629–786)	17	832 (641–1849)	7
AST^max^ (U/L)	76 (46–187)	63	49 (35–76)	19	69 (47–110)	28	206 (103–394)	16	188 (93–229)	6
ALT^max^ (U/L)	51 (33–1429)	64	41 (31–66)	18	43 (28–69)	29	156 (75–395)	17	111 (22–927)	7
GGT^max^ (U/L)	87 (43–180)	63	58 (39–140)	18	66 (3–113)	28	526 (145–988)	17	149 (44–988)	7
Total bilirubin^max^ (mg/dl)	0.8 (0.6–1.4)	64	0.7 (0.5–0.8)	18	0.7 (0.6–0.9)	29	2.5 (1.4–3.3)	17	2.8 (1.5–7.0)	7
Albumin^min^ (g/L)	26 (20–32)	58	30 (25–35)	17	28 (24–33)	28	12 (11–16)	13	16 (10–22)	7
CK^max^ (U/L)	301 (97–798)	62	161 (76–303)	19	226 (77–490)	26	1359 (768–2616)	17	480 (154–2357)	7
TNI^max^ (pg/mL)	18.7 (10.6–89.3)	61	16.8 (5.8–56.5)	19	15.8 (6.6–21.2)	25	89.3 (35.3–421)	17	70.7 (15.8–421)	7
Creatinine^max^ (mg/dL)	1.2 (0.93–2.1)	64	0.96 (0.74–1.18)	19	1.21 (0.90–1.64)	28	2.16 (1.64–3.33)	17	2.16 (1.56–4.35)	7
Hemoglobin^min^ (g/dL)	10.3 (8.0–12.5)	65	11.8 (7.6–13.6)	19	11.5 (10.3–12.6)	29	7.4 (7.0–8.4)	17	8.0 (6.7–11.4)	7
Platelet count^min^/nL	178 (134–227)	65	184 (134–233)	19	185 (142–233)	29	170 (133–183)	17	71 (49–160)	7
Absolute leukocytes^max^/nL	7.6 (5.4–9.7)	62	6.72 (4.6–8.3)	18	7.0 (5.6–9.5)	29	9.2 (7.3–11.4)	15	8.0 (6.0–17.3)	7
Absolute neutrophiles^max^/nL	5.7 (3.5–8.7)	62	4.5 (3.0–6.9)	18	5.4 (3.9–7.8)	29	8.1 (5.1–10.0)	15	5.8 (4.6–13.8)	7
Absolute lymphocytes^min^/nL	0.7 (0.5–1.0)	62	0.9 (0.6–1.4)	18	0.6 (0.5–1.0)	29	0.6 (0.5–0.8)	15	0.6 (0.4–0.9)	7
CRP^max^ (mg/lL)	172 (83–285)	65	118 (39–176)	19	121 (48–184)	29	385 (348–413)	17	348 (176–458)	7
PCT^max^ (ng/mL)	0.18 (0.05–1.20)	65	0.05 (0.02–0.24)	19	0.08 (0.05–0.24)	29	4.30 (1.0–11.0)	17	7.00.(0.77–17.0)	7
INR^max^	1.2 (1.1–1.5)	65	1.2 (1.1–1.3)	19	1.2 (1.1–1.3)	29	1.6 (1.5–2.2)	17	2.6 (1.5–3.6)	7
Fibrinogen^max^ (mg/dL)	586 (476–759)	58	518 (479–582)	16	495 (442–610)	25	855 (730–897)	17	586 (517–890)	7
Fibrinogen^min^ (mg/dL)	465 (358–531)	58	428 (287–505)	16	474 (389–561)	25	469 (364–531)	17	371 (284–453)	7
D-dimer^max^ (mg/L)	1.67 (0.86–5.08)	65	1.12 (0.51–1.93)	19	1.14 (0.69–2.27)	29	6.16 (4.36–17.23)	17	7.07 (2.24–40.24)	7

Maximum (max) or minimum (min) laboratory values of each patient during the follow up until 15 May 2020 presented as medians (interquartile ranges). Explorative comparisons of patient subgroups and corresponding *p*-values are provided in Appendix A. CK, creatine kinase; COVID-19, coronavirus disease-2019; CRP, C-reactive protein; DIC, disseminated intravascular coagulation; AST, aspartate-aminotransferase, ALT, alanine-aminotransferase; GGT, gamma-glutamyltransferase; INR, international normalized ratio; LDH, lactate dehydrogenase; PCT, procalcitonin; TNI, troponin I. ^#^ Number of total patients and patients with uncomplicated, complicated, and critical COVID-19 course, and with DIC for whom laboratory values were available. ^a^ Patients with DIC are a subset of patients with complicated and critical COVID-19.

**Table 3 jcm-10-00671-t003:** Constituting subscores of the DIC score in the seven patients with overt DIC.

Parameters of DIC Score		Points	Number of Patients (%)
**Meeting the ISTH Criteria of Overt DIC (total points ≥5) ^a^**			7 (100)
**Platelets**	>100/nL	0	3 (42.8)
	50–100/nL	1	2 (28.6)
	<50/nL	2	2 (28.6)
**D-dimer**	normal (<0.5 mg/L)	0	0 (0)
	moderately elevated (0.5–2.0 mg/L)	2	1 (14.3)
	extremely elevated (>2.0 mg/L)	3	6 (85.7)
**INR**	<1.25	0	0 (0)
	1.25–1.7	1	2 (28.6)
	>1.7	2	5 (71.4)
**Fibrinogen**	≥100 mg/dL	0	7 (100)
	<100 mg/dL	1	0 (0)

^a^ ISTH-DIC score according to Taylor et al., 2001 [26]; DIC, disseminated intravascular coagulation; INR, international normalized ratio; ISTH, International Society of Thrombosis and Haemostasis.

**Table 4 jcm-10-00671-t004:** Thrombotic microangiopathy-related factors of 22 comprehensively tested COVID-19 patients. All laboratory parameters were determined on the same day, except for the lowest platelet count.

PatID	Timepoint ^a^	Schistocytes [‰]	Haptoglobin [g/L]	Platelet Count [/nL]	Lowest ^b^ Platelet Count [/nL]	VWF Activity [%]	VWF Antigen [%]	ADAMTS13 Activity [%]	Ratio VWF:AG/ADAMTS13 Act	PCT [ng/mL]	CRP [mg/L]	Severity of COVID-19
normal	Days	<5	0.14–2.73	150–360	150–360	40–170	42–176	≥ 50	0.5–2.0 ^c^	<0.5	<5.0	
**13**	**24**	**3**	**0.22**	**309**	**173**	**n.m.**	**n.m.**	**48.0**		**1.1**	**89**	**cr (+DIC)**
29	2	Negative	2.22	208	184	170	209	64.0	3.3	0.03	7.3	uc
36	1	Negative	4.15	293	293	344	396	55.8	7.1	0.15	151	co
40	2	Negative	2.96	341	179	>390	654	75.4	8.7	0.06	31	uc
41	1	Negative	1.52	437	401	170	218	75.1	2.9	0.03	26	co
**44**	**0**	**1**	**2.55**	**85**	**85**	**>390**	**511**	**31.2**	**16.4**	**3.9**	**254**	**uc**
**44**	**1**	**5**	**2.44**	**106**	**85**	**>390**	**458**	**40.7**	**11.3**	**2.9**	**156**	**uc**
45	1	Negative	3.33	165	165	246	253	64.3	3.9	0.4	204	uc
48	2	Negative	4.25	305	275	328	409	71.6	5.7	0.12	186	co
52	1	Negative	3.51	682	597	158	218	63.5	3.4	0.04	37	co
55	14	Negative	2.96	224	193	329	221	85.4	2.6	0.12	46	cr
56	1	Negative	1.76	227	227	188	199	85.4	2.3	0.03	109	uc
**58**	**1**	**Negative**	**1.79**	**250**	**200**	**>390**	**782**	**46.0**	**17.0**	**3.5**	**184**	**co**
59	0	9	3.31	240	213	336	226	88.5	2.6	0.02	32	uc
70	9	Negative	1.44	392	187	215	228	80.0	2.9	0.05	15	uc
73	0	Negative	1.71	254	254	178	202	97.8	2.1	<0.02	0.79	uc
**74**	**44**	**Negative**	**2.52**	**54**	**54**	**>390**	**595**	**17.8**	**33.4**	**6.8**	**325**	**cr (+DIC)**
77	15	Negative	3.81	245	119	>390	613	90.9	6.7	0.12	32	uc
78	1	Negative	3.51	186	159	361	222	85.4	2.6	0.23	179	uc
79+	1	Negative	5.45	220	49	>390	602	76.4	7.9	0.31	155	cr (+DIC)
81+	0	n. m.	n.m.	345	160	267	234	87.0	2.7	0.28	169	cr (+DIC)
84	0	Negative	4.21	759	759	263	231	71.8	3.2	0.24	138	uc
85	1	Negative	3.30	238	238	162	195	84.8	2.3	0.03	39	Uc

Data of patients with decreased ADAMTS13 activity are in bold. CRP, C-reactive protein; COVID-19, coronavirus disease-19; DIC, disseminated intravascular coagulation; PatID, patient identification number; PCT, procalcitonin; uc, uncomplicated; co, complicated; cr, critical COVID-19 course; n. m., not measured; VWF:AG, von Willebrand-factor antigen. ^a^ Day when TMA-related factors were assayed after hospitalisation. ^b^ Lowest platelet count of each patient during hospitalization. ^c^ In healthy normal controls the ratio of VWF:Ag/ADAMTS13 act is usually between about 0.5 and 2.0, **+** Patients who died from COVID-19 during hospitalization.

## Data Availability

The data presented in this study are available on request from the corresponding author. The data are not publicly available due to ethical and privacy restrictions.

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
