# Peer review of "No Evidence for Classic Thrombotic Microangiopathy in COVID-19"

_jcm, 2021, doi:10.3390/jcm10040671_

Round 1

Reviewer 1 Report

The authors address the open question on the pathophysiology of the COVID-19-disease associated coagulopathy and the histologically described phenomenon of thrombotic microangiopathy in the lung (and to a lesser extent in the kidney). In their small cohort of patient (n=65) systemic signs of TMA (MAHA and thombocytopenia) were more or less absent and ADAMTS13 activity was mainly within normal limits - although haptoglobin and ADAMTS13 levels were only measured in 22 patients (!) The authors point to a dysbalance between high VWF concentrations and low levels regulating proteases (ADAMTS13) and at least cite the deposition of complement activation products as sign of activated complement pathways. Although the conclusions from their results are clear, I would like to ask the authors for a more detailed description of the pathophysiology of the different TMA forms, a more detailed discussion of the limitations of their study and the discussion of potentially involved mechanisms. One of the clear limitations of their study is the total lack of measurement of complement activation in the blood.

Introduction: 

Please explain TMA and subgroups with their different pathophysiology (ADAMTS13 / alternative complement activation / other) in more detail in the introduction as well as the discussion section. For the discussion I would also ask to add a descritpion of well knomwn localiced TMA forms e.g. in the kidney without systemic signs of TMA.

Materials and Methods

Please add a statement on ADAMTS13 measurement.

Results:

Text and Table 1: please do not use different definitions of obesity or define both (... obesit could be determined in 62/65 patients, obesity >30 ...)

Tables and Figures: please explain abbreviations throughout

3.2 and Table 2:

Please consider for the kideny function a different wording than: "elevated serum creatinine ... indicated impaired renal function". (? At hospital admission ... patients presented with ... During ...  XX patients received renal replacement therapy ..).

Table 2 is rather long and confusing. Please consider to put part of this table in the supplemenary files.

3.3.3 Patients with DIC:

The authors present data on 65 patients. Why the phrase: ... in whom the DIC score could be assessed. ?

Figure 2: The correct headline is missing.

3.3.4 and Table 4:

For schistocytes please use the same unit: % or °/°°.

Further comment: "TMA-related factors" including haptoglobin were only measured in 22 patients - and in only 4 out of 7 DIC patients (the other 3 with thrombocytopenia). Furthermore the day of this measurement is not the day of the lowest thrombocyte count. Again complement diagnostic is lacking at all. I would suggest at least to add the day after hospital admission - so that it becomes clear when these measurements were done during the disease course. 

  Unfortunately, the literature on COVID-19-disease is evolving fast, please consider to discuss the following literature:

  • Diorio et al. Evidence of thrombotic microangiopathy in children with SARS-CoV-2 across the spectrum of clinical presentations. Blood Adv. 2020 Dec 8;4(23):6051-6063. doi:10.1182/bloodadvances.2020003471.
  • Henry et al. ADAMTS13 activity to von Willebrand factor antigen ratio predicts acute kidney injury in patients with COVID-19: Evidence of SARS-CoV-2 induced secondary thrombotic microangiopathy. Int J Lab Hematol. 2020 Dec 3;10.1111/ijlh.13415. doi: 10.1111/ijlh.13415. Online ahead of print.

  • Beigee et al. Diffuse alveolar damage and thrombotic microangiopathy are the main histopathological findings in lung tissue biopsy samples of COVID-19 patients. Pathol Res Pract. 2020 Oct;216(10):153228. doi:10.1016/j.prp.2020.153228. Epub 2020 Sep 19.

  • Mastellos et al. Complement C3 vs C5 inhibition in severe COVID-19: Early clinical findings reveal differential biological efficacy. Clin Immunol. 2020 Nov;220:108598. doi:10.1016/j.clim.2020.108598. Epub 2020 Sep 19.

Reviewer 2 Report

Overall, the article is well written. The authors made an attempt to evaluate clinical and laboratory parameters in German COVID-19 patients.
Despite this, I have a few comments that the authors must pay attention to.
Introduction:
1. Please expand the SARS abbreviation.
2. Line 46 - I would replace coagulopathy with hemostasis or blood coagulaton.
3. The incidence of thromboembolic complications in COVID-19 is higher than that reported by the authors, please correct this. In some studies, it was 23-49%.
4. In the introduction, the authors mention DIC - I agree that preliminary studies showed a very high incidence of this complication in COVID-19, however, later manuscripts showed that it is not a classic form of DIC, and patients do not meet the criteria for diagnosing DIC. Please, refer to it.
5. Please explain the abbreviation vWF and ADAMS13. Maybe it doesn't make sense to describe these proteins in such detail. Maybe it is better to write in general that hemostatic disorders develop as a result of several overlapping processes ...
7. Generally, I suggested to add the sentence that the endothelium is also damaged in the course of COVID-19.
8. The aim of the study should be described more precisely.
Results:
1. Please add the p-value to table 1 and table 2.
2. All abbreviations in table 1 and table 2 should be explained.
3. The authors should explain why obesity was not assessed in 3 patients - is it not possible on the basis of BMI?
4. The title of sub-item 3.2 corresponds only partially to the content of Table 2.
5. Fig. 2 should be precisely described in the text of the manuscript.

In the discussion, I would add the limitations of your research.

Above all, my remarks do not detract from the great scientific and practical value of the manuscript.

Round 2

Reviewer 2 Report

The authors have satisfactorily addressed all of my comments. My last note to the authors is to discuss their own observations in the context of a recent work showing hypercoagulation in the absence of fibrinolysis in COVID-19 (Thromb Haemost. 2021 Jan 5. doi: 10.1055/a-1346-3178. Online ahead of print.).

Congratulation and Happy New Year!

Author Response

Dear Referees,

Thank you for the additional comments, which reflect the fast evolving field of COVID-19. Following your suggestions, we have added additional citations to the manuscript and provided a brief comment within the text. These references will definitely provide readers with additional insights into this topic.

Please find the following minor revisions as follows:

After mentioning atypical DIC during COVID-19 we referred to the ongoing discussion of this topic by citing as follows (Page 12):

This data resembles observations by other groups (20, 37) and new designations for the COVID-19 associated hemostatic disturbances have been proposed, i.e. “COVID-19 asso-ciated coagulopathy (CAC)” (18) or “Pulmonary intravascular coagulopathy (PIC)” (43) to stress the discrepancy to “classic DIC” (44). Experts in this field have discussed the current evidence and suggested that CAC during severe COVD-19 should be considered as a prothrombotic phenotype of DIC (45)

Iba, T., et al. Proposal of the definition for COVID-19-associated coagulopathy. J Clin Med 2021 ;10,191

We further mentioned the potential of viscoelastic methods to identify patients with hemostatic abnormalities in the paragraph: 4.1 Limitations of the study (Page 13).

In addition, viscoelastic methods to study clot formation and fibrinolysis were not performed in this COVID-19 cohort, although these methods could be useful to identify the thrombotic risk in individual patients and/or to guide antithrombotic treatment (58).

Slomka, A., et al. Haemostasis in coronavirus disease 2019 – lesson from viscoelastic methods : a systematic review. Thromb Haemost 2021. Doi :10.1055/a-1346-3178

With kind regards,

Martin Sprinzl